# Effectiveness of D-dimer in predicting distant metastasis in colorectal cancer

Xin Zhang[1], Wenxing Li[2], Xuan Wang[1], Jinhe Lin[1], Chengxue Dang[1], Dongmei Diao[1]*

**1** Department of Oncology Surgery, First Affiliated Hospital of Xi'an Jiaotong University, Xi'an, Shaanxi, China, **2** Department of Radiotherapy, Oncology Department, The First Affiliated Hospital of Xi 'an Jiaotong University, Xi'an, Shaanxi, China

\* diaomei310@xjtu.edu.cn

## Abstract

### Purpose

Patients with cancer often present with a hypercoagulable state, which is closely associated with tumor progression. The purpose of this study was to assess the diagnostic efficacy of D-dimer in predicting distant metastasis in colorectal cancer (CRC).

### Methods

This study included 529 patients diagnosed with CRC at our hospital between January 2020 and December 2022. Plasma coagulation indicators and tumor markers were collected prior to treatment and their diagnostic efficacy for predicting CRC metastasis was assessed by receiver operating characteristic (ROC) curves. Independent risk factors for evaluating tumor metastasis were obtained by multivariate logistic regression analysis.

### Results

The level of D-dimer in the metastatic group was significantly higher than that in the non-metastatic group (*P*<0.001). The results of the multiple logistic regression analysis indicated that lower level of prealbumin and platelet, and higher level of glucose, CEA and D-dimer were independent risk factors for distant metastasis in patients with CRC (P<0.05, respectively). The combination of prealbumin, glucose, D-dimer, platelet and tumor markers (PRE2) was found to be significantly more effective in predicting metastasis of CRC when compared to the combination of tumor marker alone (PRE1, P<0.001).

### Conclusion

Plasma D-dimer may be a novel tumor marker for screening metastases of CRC.

**Competing interests:** The authors have declared that no competing interests exist.

## Introduction

CRC is the third most common tumor globally and the second leading cause of tumor-related death [1]. In recent years, with advancements in surgical techniques, perioperative management, and chemotherapy, the survival rates of patients with CRC have improved. But there are still many patients who have metastases when they are first diagnosed, and metastatic CRC remains incurable in most cases [2]. Therefore, it is crucial to identify metastasis of CRC in its early stages.

Solid malignant tumors are frequently linked to abnormal coagulation function, specifically hypercoagulation and hyperfibrinolysis [3, 4]. D-dimer and fibrinogen degradation products (FDP) are protein fragments released into the systemic circulation due to degradation of blood clots. Many coagulation abnormalities, such as thrombocytosis, high fibrinogenemia, elevated D-dimer levels and FDP, are associated with the progression or poor prognosis of CRC [5–7]. In addition, many studies have found that the hypercoagulable state of blood is closely related to the metastasis and development of tumors [8, 9]. Elevated plasma D-dimer levels are associated with poor prognosis in CRC [10]. Emerging evidence also suggested that plasma D-dimer levels were progressively increased in healthy people, non-metastatic non-small cell lung cancer (NSCLC) patients, and patients with metastatic NSCLC [11]. Tumor cells promote the hypercoagulable state by secreting tissue factors and procoagulant substances, and interact with platelets to form microthrombus and promote hematogenous metastasis. Targeting tumor-associated tissue factors can effectively prevent metastasis and tumor-associated hypercoagulability [12].

These results suggest that coagulation markers may have the potential to predict tumor metastasis. In fact, coagulation-related indicators fibrinogen, D-dimer, and activated partial prothrombin time (APTT) are independent risk factors for NSCLC bone metastasis [13]. Elevated D-dimer has also been found to be associated with an increased number of circulating tumor cells in patients with advanced breast cancer [14]. Our previous studies have shown that D-dimer and FDP have a diagnostic performance comparable to gastric cancer markers in predicting gastric cancer metastasis. However, few studies have reported the predictive value of D-dimer for distant metastases of CRC.

## Materials and methods

### Study subjects

A total of 529 CRC patients who visited the First Affiliated Hospital of Xi'an Jiaotong University between January 2020 and December 2022 were retrospectively enrolled in the study. We began collecting medical records on March 1, 2023, and during or after data collection, we were able to obtain information that would identify individual participants. Patients were included if they met the following criteria: (a) confirmed diagnosis by imaging or pathological specimen, (b) first diagnosis, and (c) all patients were tested for coagulation and tumor markers. Patients who met any of the following conditions were excluded: (a) acute infection, (b) had a coagulation system disease or took anticoagulants, (c) had other malignant diseases, or (d) had a history of chemoradiotherapy. Data was collected for CRC patients who had not received any previous treatment and were grouped based on their metastatic status. This study has been approved by the Ethics Committee of the First Affiliated Hospital of Xi'an Jiaotong University. Informed consent was obtained from all subjects and/or their legal guardian(s).

## Statistical analysis

Gender data is expressed as frequency and percentage and compared using a chi-square test. Continuous data was tested for normality, with normally distributed data represented by means and standard deviations, and differences were compared using t-tests. Non-normal distribution data was represented by medians and interquartile ranges and statistically analyzed using the Mann-Whitney U test. The performance of laboratory indicators in predicting CRC metastasis was evaluated using ROC curve analysis. The Youden Index was obtained by the formula sensitivity+specificity-1. Cut-off values, along with sensitivity and specificity, were obtained based on the maximum value of the Youden Index. DeLong's test was used to compare ROC curves. The continuous variable was converted into a two-component variable based on the cut-off value. Multivariate logistic regression analysis was performed to identify independent risk factors for distant metastases in CRC. All statistical analyses, except for the comparison of ROC curves using MedCalc (version 19.4.1), were conducted using SPSS (version 25.0). A two-tailed $P$-value less than 0.05 was considered statistically significant.

## Results

### Plasma D-dimer level was markedly elevated in metastatic CRC

Of the 529 patients enrolled with CRC, 193 had distant metastases. As shown in Table 1, there were no significant differences in gender distribution, age, APTT and thrombin time (TT). The metastatic group exhibited lower level of albumin (median: 37.64 vs 40.4), prealbumin (median: 168.79 vs 227.15), prothrombin activity (PTA, median: 90 vs 93.95, $P = 0.001$) and platelet counts (median: 181 vs 218.5) and higher level of globulin (median: 27.45 vs 26.8,

**Table 1. Demographic and baseline characteristics of patients (N = 529).**

| Characteristic | Total (N = 529) | Non-metastasis (N = 336) | Metastasis (N = 193) | P value |
|---|---|---|---|---|
| Gender (Female) | 181(34.2) | 125(37.2) | 56(29) | 0.056 |
| Age, year | 60(52–67) | 60(52–67.75) | 60(52–66) | 0.509 |
| Albumin, g/l | 39.55(36.1–42.6) | 40.4(37.2–43) | 37.64(33.89–41.48) | <0.001 |
| Globulin, g/l | 27.2(24.5–30.2) | 26.8(24.38–29.7) | 27.45(24.72–31.08) | 0.03 |
| Prealbumin, mg/l | 211(164.05–255.81) | 227.15(189.85–267.78) | 168.79(109.77–218.4) | <0.001 |
| Glucose, mmol/l | 4.67(4.21–5.27) | 4.57(4.15–5.12) | 4.76(4.34–5.55) | 0.008 |
| CEA, ng/ml | 4.37(2.12–14.13) | 3.75(2.02–9.37) | 5.58(2.42–35.97) | <0.001 |
| CA199, U/ml | 13.82(7.76–29.99) | 11.65(7.22–23.22) | 21.27(8.94–82.5) | <0.001 |
| PT, second | 13.2(12.8–13.8) | 13.2(12.7–13.6) | 13.4(12.9–14) | <0.001 |
| PTA, % | 92.9(81.15–102.45) | 93.95(84.35–103.55) | 90(75.35–100.2) | 0.001 |
| PTR | 1.03(0.99–1.08) | 1.03(0.98–1.07) | 1.05(1.01–1.12) | <0.001 |
| INR | 1.03(0.98–1.08) | 1.02(0.97–1.06) | 1.05(1–1.11) | <0.001 |
| APTT, second | 35.8(33.2–38.6) | 35.7(33.3–38.4) | 35.95(33.13–38.88) | 0.726 |
| TT, second | 16.4(15.7–17) | 16.4(15.8–17) | 16.3(15.4–17) | 0.088 |
| Fibrinogen, g/l | 3.33(2.78–4.14) | 3.27(2.76–3.93) | 3.58(2.79–4.4) | 0.011 |
| D-dimer, mg/l | 0.7(0.47–1.5) | 0.6(0.4–0.9) | 1.5(0.6–3.4) | <0.001 |
| FDP, mg/l | 1.9(1.23–3.9) | 1.7(1.1–2.6) | 3.35(1.6–8.3) | <0.001 |
| Platelet, 10^9/l | 210(155–264) | 218.5(169–273) | 181(133–250) | <0.001 |

Values in parentheses are percent or interquartile ranges. Abbreviations: CEA, carcinoembryonic antigen; CA199, carbohydrate antigen 199; CA724, carbohydrate antigen 724; PT, prothrombin time; PTA, prothrombin activity; PTR, prothrombin ratio; INR, international normalized ratio; APTT, activated partial prothrombin time; TT, thrombin time; FDP, fibrin degradation products.

$P$ = 0.03), blood glucose (median: 4.76 vs 4.57, $P$ = 0.008), carcinoembryonic antigen (CEA, median: 5.58 vs 3.75), carbohydrate antigen 199 (CA199, median: 21.27 vs 11.65) and D-dimer (median: 1.5 vs 0.6) than non-metastatic group (unless otherwise specified, $P<0.001$). The levels of PT, PTR, INR, fibrinogen and FDP also elevated in the metastatic group (Table 1, all $P<0.05$).

## Plasma D-dimer level had higher diagnostic efficacy than conventional markers for CRC metastasis

We conducted ROC analysis on the differential parameters between the two groups mentioned above to assess the diagnostic efficacy of each index. The area under the curve (AUC) of albumin was 0.637 with a 95% confidence interval (CI) of 0.586–0.688. Using the cutoff value of 38.77, the sensitivity and specificity of albumin were 0.65 and 0.594, respectively. The AUC (95% CI) of prealbumin was 0.739 (0.691–0.786), with a sensitivity of 0.793 and a specificity of 0.607 at the cutoff value of 184.205. The AUC (95% CI) of glucose was 0.569 (0.518–0.62), and when the cutoff value was 5.295 mmol/l, the sensitivity and specificity were 0.328 and 0.802, respectively. The AUC (95% CI) of platelet was 0.604 (0.553–0.656) with a sensitivity of 0.664 and a specificity of 0.524 at the cutoff value of 186.5, of D-dimer was 0.738 (0.69–0.785) with a sensitivity of 0.583 and a specificity of 0.836 at the cutoff value of 1.125 mg/l, of FDP was 0.706 (0.657–0.754) with a sensitivity of 0.625 and a specificity of 0.722 at the cutoff value of 2.35 mg/l. As for tumor markers, the AUC (95% CI) of CEA was 0.608 (0.553–0.663). When the cutoff value of CEA was 14.465 U/ml, the sensitivity was 0.38 and specificity was 0.825. The AUC (95% CI) of CA199 was 0.641 (0.585–0.697) with a sensitivity of 0.361 and a specificity of 0.891 at the cutoff value of 41.58 U/ml (Table 2 and Figs 1 and 2). The Youden index, positive predictive value (PPV) and negative predictive value (NPV) were also shown in Table 2.

To identify independent risk factors for CRC metastasis, logistic regression analysis was performed. The univariate analysis showed that higher globulin, glucose, CEA, CA199, PT, PTR, INR, fibrinogen, D-dimer and FDP, and lower levels of albumin, prealbumin, PTA and platelet were associated with an increased risk of CRC metastasis. Then multivariate logistic

**Table 2. Diagnostic performance of parameters for predicting CRC metastases.**

| Item | AUC | 95% CI | Cut-off | Sen | Spe | Youden index | NPV | PPV | P value |
|------|-----|--------|---------|-----|-----|--------------|-----|-----|---------|
| Albumin | 0.637 | 0.586–0.688 | 38.77 | 0.650 | 0.594 | 0.243 | 0.506 | 0.264 | <0.001 |
| Globulin | 0.557 | 0.504–0.609 | 32.705 | 0.208 | 0.919 | 0.127 | 0.669 | 0.597 | 0.03 |
| Prealbumin | 0.739 | 0.691–0.786 | 184.205 | 0.793 | 0.607 | 0.401 | 0.404 | 0.199 | <0.001 |
| Glucose | 0.569 | 0.518–0.62 | 5.295 | 0.328 | 0.802 | 0.131 | 0.675 | 0.488 | 0.008 |
| CEA | 0.608 | 0.553–0.663 | 14.465 | 0.380 | 0.825 | 0.205 | 0.708 | 0.543 | <0.001 |
| CA199 | 0.641 | 0.585–0.697 | 41.58 | 0.361 | 0.891 | 0.252 | 0.732 | 0.629 | <0.001 |
| PT | 0.598 | 0.546–0.649 | 13.68 | 0.396 | 0.788 | 0.184 | 0.695 | 0.517 | <0.001 |
| PTA | 0.593 | 0.539–0.647 | 76.95 | 0.883 | 0.284 | 0.167 | 0.442 | 0.297 | 0.001 |
| PTR | 0.593 | 0.543–0.644 | 1.105 | 0.271 | 0.88 | 0.151 | 0.677 | 0.565 | <0.001 |
| INR | 0.626 | 0.576–0.676 | 1.065 | 0.427 | 0.79 | 0.218 | 0.706 | 0.539 | <0.001 |
| Fibrinogen | 0.566 | 0.513–0.619 | 4.16 | 0.349 | 0.818 | 0.167 | 0.687 | 0.523 | 0.011 |
| D-dimer | 0.738 | 0.69–0.785 | 1.125 | 0.583 | 0.836 | 0.419 | 0.778 | 0.67 | <0.001 |
| FDP | 0.706 | 0.657–0.754 | 2.35 | 0.625 | 0.722 | 0.347 | 0.771 | 0.563 | <0.001 |
| Platelet | 0.604 | 0.553–0.656 | 186.5 | 0.664 | 0.524 | 0.188 | 0.536 | 0.285 | <0.001 |

Abbreviations: AUC, area under receiver operating characteristics; CI, confidence interval; Sen, sensitivity; Spe, specificity; PPV, positive predictive value; NPV, negative predictive value.

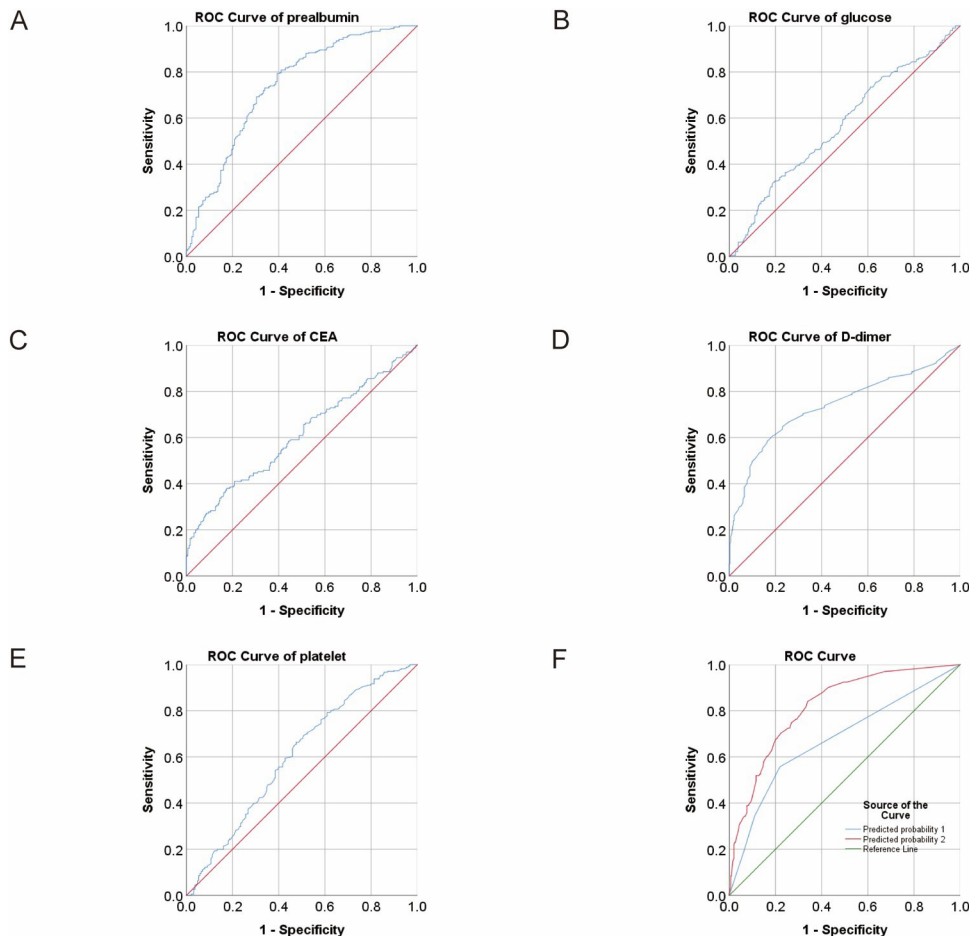

**Fig 1. ROC analysis for the prediction of CRC metastasis.** AUC indicates the diagnostic power of prealbumin, glucose, CEA, D-dimer, platelet and predicted probability (PRE1 and PRE2).

regression analysis was conducted and the results indicated that prealbumin [odds ratio (OR): 0.458, 95% CI: 0.239–0.874, *P* = 0.018], glucose (OR: 2.014, 95% CI: 1.095–3.705, *P* = 0.024), CEA (OR: 1.955, 95% CI: 1.082–3.532, *P* = 0.026), D-dimer (OR: 3.104, 95% CI: 1.446–6.663, P = 0.004) and platelet (OR: 0.574, 95% CI: 0.331–0.994, *P* = 0.0448) were independent markers for CRC metastasis (Table 3).

In order to clarify the improvement of the above risk factors on the prediction efficiency of metastasis, we conducted a combined ROC analysis. The combination of prealbumin, glucose, D-dimer, platelet, CEA and CA199 (PRE2) was significantly more effective in predicting CRC metastasis than the combination of CEA and CA199 (PRE1) alone (P<0.001). The AUC (95% CI) of PRE2 was 0.817 (0.775–0.859) with a sensitivity of 0.84 and a specificity of 0.661 (Table 4 and Fig 1).

## Discussion

Our study focuses on exploring the connection between coagulation and the metastatic status of CRC. Our results indicate that a combination of multiple indicators is crucial in assessing tumor metastasis in CRC patients who have recently been diagnosed. These findings are particularly useful in clinical evaluations of tumor metastasis in CRC patients.

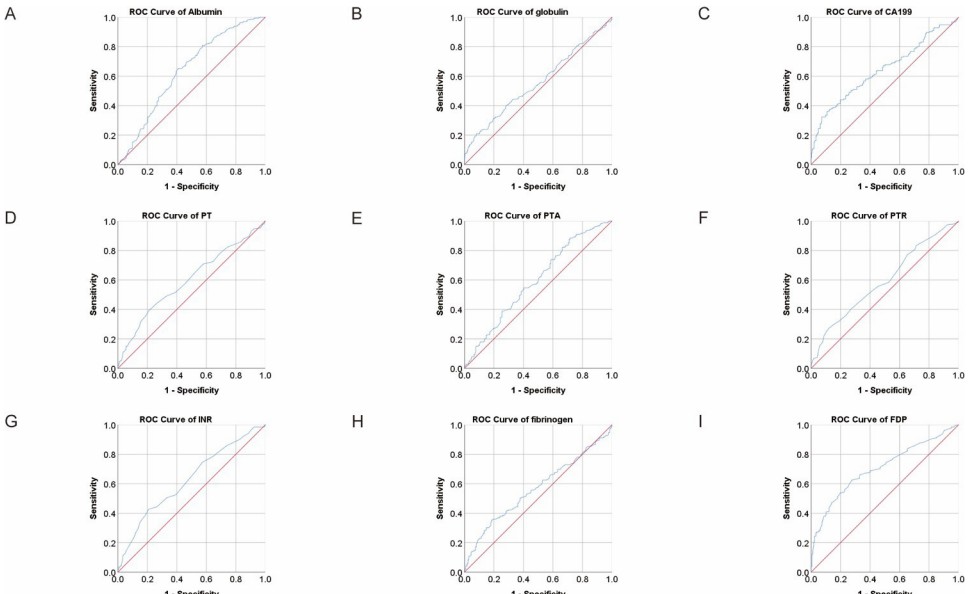

**Fig 2. ROC analysis for the prediction of CRC metastasis.** AUC indicates the diagnostic power of albumin, globulin, CA199, PT, PTA, PTR, INR, fibrinogen and FDP.

The metastasis of tumor cells is the primary cause of death, thus making the early detection of tumor metastasis especially crucial. Cancer spread is a complex process that involves multiple steps. Initially, tumor cells break away from the primary tumor and spontaneously metastasize. To accomplish effective metastasis, these cells must survive in blood vessels, overcome the mechanical flow of blood and immune system components, undergo trans-endothelial migration, and colonize within distant vascular beds. Establishment of a new supportive vasculature

**Table 3. Binary logistic regression analyses of variables for CRC metastasis.**

| Item | Univariate analysis | | | Multivariate analysis | | |
|---|---|---|---|---|---|---|
| | Odds ratio | 95% CI | P value | Odds ratio | 95% CI | P value |
| Albumin | 0.369 | 0.256–0.532 | <0.001 | 0.862 | 0.467–1.593 | 0.637 |
| Globulin | 2.992 | 1.769–5.06 | <0.001 | 1.422 | 0.622–3.251 | 0.404 |
| Prealbumin | 0.168 | 0.112–0.253 | <0.001 | 0.458 | 0.239–0.874 | 0.018* |
| Glucose | 1.983 | 1.324–2.971 | 0.001 | 2.014 | 1.095–3.705 | 0.024* |
| CEA | 2.885 | 1.874–4.441 | <0.001 | 1.955 | 1.082–3.532 | 0.026* |
| CA199 | 4.628 | 2.841–7.539 | <0.001 | 1.918 | 0.981–3.751 | 0.057 |
| PT | 2.436 | 1.649–3.599 | <0.001 | 0.226 | 0.044–1.16 | 0.075 |
| PTA | 0.335 | 0.208–0.539 | <0.001 | 0.438 | 0.157–1.227 | 0.116 |
| PTR | 2.73 | 1.726–4.319 | <0.001 | 0.503 | 0.152–1.656 | 0.258 |
| INR | 2.811 | 1.906–4.148 | <0.001 | 4.316 | 0.875–21.293 | 0.073 |
| Fibrinogen | 2.408 | 1.604–3.614 | <0.001 | 1.099 | 0.568–2.128 | 0.780 |
| D-dimer | 7.127 | 4.743–10.71 | <0.001 | 3.104 | 1.446–6.663 | 0.004* |
| FDP | 4.337 | 2.973–6.326 | <0.001 | 1.491 | 0.698–3.188 | 0.303 |
| Platelet | 0.46 | 0.319–0.663 | <0.001 | 0.574 | 0.331–0.994 | 0.048* |

The reference for each parameter is a range that is less than their respective cut-off values shown in Table 2.

*$P<0.05$.

**Table 4. Comparison of ROC curves.**

| Item | AUC | 95% CI | Cut-off | Sen | Spe | Youden index | *P* value[1] |
|------|-----|--------|---------|-----|-----|--------------|-----------|
| PRE1 | 0.676 | 0.618–0.734 | 0.316 | 0.557 | 0.781 | 0.338 | Reference |
| PRE2 | 0.817 | 0.775–0.859 | 0.216 | 0.840 | 0.661 | 0.501 | <0.001 |

Abbreviations: ROC, receiver operating characteristic; AUC, area under receiver operating characteristics; CI, confidence interval; PRE1, prediction probability obtained by binary logistic regression combining CEA and CA199; PRE2, prediction probability obtained by binary logistic regression combining prealbumin, glucose, CEA, D-dimer and platelet.

[1]The P value was obtained by the DeLong's test.

and growth are the main challenges that tumor cells must face [15]. The activation of coagulation is known to promote hematogenous tumor metastasis through several mechanisms. Specifically, a clot comprising of platelets, fibrinogen, and tumor cells can enhance adhesion to the endothelium [16], as well as assist tumor cells in avoiding immune surveillance, thus facilitating the spread of cancer [17]. Additionally, platelets can promote epithelial-mesenchymal transformation (EMT) in tumor cells by activating the TGFβ/SMAD and NF-κB pathways [18]. In metastasis, the coagulation process initiated by tissue factor, fibrinogen, and protease-activated receptor-4 (PAR4) signaling activates platelets, which in turn promote tumor cell survival in the bloodstream and facilitate distant metastasis [19]. Thrombin was found to promote colon cancer proliferation and tumor growth through PAR-1 and ECM protein fibrinogen. The study showed that tumor growth and proliferation decreased in mice deficient in PAR-1 and fibrinogen [20].

Anticoagulant or antithrombotic therapy can improve outcomes for cancer patients. Low molecular weight heparin (LMWH) can prolong survival in cancer patients with or without deep vein thrombosis [21]. Not only this, LMWH tinzaparin inhibits adhesion and invasion of human pancreatic tumor cells [22]. Furthermore, Hirudin functions as a highly specific thrombin inhibitor, effectively suppressing tumor metastasis and therefore prolonging survival [23].

A significant body of evidence suggests that the activation of coagulation plays a pivotal role in the promotion of tumor metastasis. Hence, it can be inferred that coagulation indicators hold the potential to serve as predictors for metastasis. In the case of gastric cancer patients, it has been observed that their plasma D-dimer levels were significantly higher when compared to healthy controls [24]. In NSCLC, with tumor formation and tumor metastasis, the level of D-dimer also gradually increased [11]. Fibrinogen is known to promote gallbladder cancer cell metastasis by inducing the expression of intercellular adhesion molecule 1 (ICAM1) [25]. Our previous studies have confirmed the value of D-dimer and FDP in predicting distant metastasis of gastric cancer [26]. APTT primarily reflects the body's endogenous coagulation function, and a reduced level indicates a state of hypercoagulation. This may lead to thrombotic diseases. D-dimer, on the other hand, reflects fibrinolytic function, and an increased level typically signifies hypercoagulability, secondary hyperfibrinolysis, disseminated intravascular coagulation, and other related diseases [27]. Based on our findings, significant differences in coagulation indexes were observed between the CRC metastasis group and the control group, leading us to hypothesize that CRC patients in a hypercoagulable state are more prone to developing distant metastases. Our results showed that patients with distant metastatic CRC have significantly elevated plasma D-dimer levels compared with primary CRC. We obtained the diagnostic parameters and cut-off values for each index through ROC analysis. Further, multivariate logistic regression confirmed that lower level of prealbumin and platelet, and higher level of glucose, CEA and D-dimer were independent risk factors for CRC distant metastasis. Combined ROC analysis confirmed that the combination of all risk factors had higher diagnostic

efficiency than only combined CEA and CA199. This suggests that D-dimer may serve as novel tumor markers for predicting distant metastases in CRC, provided that other disorders that cause abnormally elevated coagulation indicators are excluded.

In our hospital, the level of D-dimer used to assess normal coagulation function was considered to be less than 1 mg/L. False-positives for D-dimer values are frequent in older patients, patients with cancer or systemic infection, pregnancy, recent surgery, or trauma [28]. It is essential to interpret the clotting indicators in conjunction with other clinical findings and imaging tests to avoid misdiagnosis and unnecessary treatment. The retrospective study conducted at a single center and the small sample size of participants somewhat limit the scope and generalizability of this study. In addition, the P value of platelets in multivariate analysis is closer to 0.05, and more data are needed to determine whether it is a risk factor for CRC metastasis.

To conclude, plasma D-dimer had higher diagnostic performance than traditional tumor markers in the evaluation of CRC metastasis. Combining D-dimer with CEA and CA199 provides a more accurate assessment of CRC metastatic status.

## Supporting information

**S1 Data.**
(XLSX)

## Author Contributions

**Conceptualization:** Chengxue Dang, Dongmei Diao.

**Data curation:** Xin Zhang, Wenxing Li, Xuan Wang.

**Formal analysis:** Xin Zhang, Wenxing Li, Xuan Wang.

**Investigation:** Xin Zhang, Jinhe Lin.

**Methodology:** Xin Zhang, Jinhe Lin, Chengxue Dang, Dongmei Diao.

**Project administration:** Wenxing Li, Chengxue Dang, Dongmei Diao.

**Resources:** Xin Zhang, Xuan Wang, Jinhe Lin.

**Software:** Xin Zhang, Wenxing Li, Xuan Wang.

**Supervision:** Chengxue Dang.

**Validation:** Dongmei Diao.

**Visualization:** Chengxue Dang, Dongmei Diao.

**Writing – original draft:** Xin Zhang, Xuan Wang.

**Writing – review & editing:** Xin Zhang, Chengxue Dang, Dongmei Diao.

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
