## [Decision Letter · Decision Letter 0]

27 May 2024

PONE-D-23-43532Effectiveness of D-dimer in predicting distant metastasis in colorectal cancerPLOS ONE

Dear Dr. zhang,

Thank you for submitting your manuscript to PLOS ONE. After careful consideration, we feel that it has merit but does not fully meet PLOS ONE’s publication criteria as it currently stands. Therefore, we invite you to submit a revised version of the manuscript that addresses the points raised during the review process.

The manuscript is substantially well written. Only minor revisions are needed.

We look forward to receiving your revised manuscript.

Kind regards,

Raffaele Serra, M.D., Ph.D

Academic Editor

PLOS ONE

2. In the online submission form, you indicated that [All relevant data are available from the corresponding author.]. 

Additional Editor Comments:

Only minor revisions are required.

Reviewers' comments:

Reviewer's Responses to Questions

**Comments to the Author**

1. Is the manuscript technically sound, and do the data support the conclusions?

Reviewer #1: Yes

2. Has the statistical analysis been performed appropriately and rigorously? 

Reviewer #1: I Don't Know

3. Have the authors made all data underlying the findings in their manuscript fully available?

Reviewer #1: No

4. Is the manuscript presented in an intelligible fashion and written in standard English?

Reviewer #1: Yes

5. Review Comments to the Author

Reviewer #1: - Can you explain the fact that sensibility and specificity values of PT, PTA, PTR and INR are so different? They are different ways of the same test, and we do not expect such different results.

- You say: "The levels of PT, PTR, INR, fibrinogen and FDP also elevated in the metastatic group (Table 1, all P<0.05)." I understand that in these tests p-value is between 0.001 and 0.05, but in Table 1, it is marked these p-values are <0.001. Indicate which are the correct p-values.

6. PLOS authors have the option to publish the peer review history of their article (what does this mean?). If published, this will include your full peer review and any attached files.

Reviewer #1: No

---

## [Author Response · Author response to Decision Letter 0]

17 Jun 2024

Dear editor,

Thanks for your comments and suggestions. We have revised our manuscript according to the reviewers' comments, as follows:

Our response: We have made changes to the manuscript according to PLOS ONE's style requirements. In line 7, We changed the font size of ‘Abstract’ to 18pt. We also changed the parentheses of the citations to the middle parentheses.

2. In the online submission form, you indicated that [All relevant data are available from the corresponding author.].

Our response: We have added data to the additional information.

Our response: After careful examination, we did not cite the retracted papers.

Additional Editor Comments:

Only minor revisions are required.

Reviewers' comments:

Reviewer's Responses to Questions

Comments to the Author

1. Is the manuscript technically sound, and do the data support the conclusions?

Reviewer #1: Yes

2. Has the statistical analysis been performed appropriately and rigorously?

Reviewer #1: I Don't Know

3. Have the authors made all data underlying the findings in their manuscript fully available?

Reviewer #1: No

4. Is the manuscript presented in an intelligible fashion and written in standard English?

Reviewer #1: Yes

5. Review Comments to the Author

Reviewer #1: - Can you explain the fact that sensibility and specificity values of PT, PTA, PTR and INR are so different? They are different ways of the same test, and we do not expect such different results.

Our response: Dear reviewer, thank you for your valuable comments. As you said, PT, PTA and PTR are different ways of the same test. It is not difficult to find that they have a similar AUC (95% CI) and Youden Index. Since we determine the cut-off value based on the optimal Youden index, the difference of cut-off value will inevitably cause the difference in sensitivity and specificity, but we can find that their diagnostic efficiency is similar.

- You say: "The levels of PT, PTR, INR, fibrinogen and FDP also elevated in the metastatic group (Table 1, all P<0.05)." I understand that in these tests p-value is between 0.001 and 0.05, but in Table 1, it is marked these p-values are <0.001. Indicate which are the correct p-values.

 Our response: Dear reviewer, all P values are correct, and the P<0.05 we annotated in the text contains P<0.001.

6. PLOS authors have the option to publish the peer review history of their article (what does this mean?). If published, this will include your full peer review and any attached files.

Do you want your identity to be public for this peer review? For information about this choice, including consent withdrawal, please see our Privacy Policy.

Reviewer #1: No

---

## [Editor Report · Decision Letter 1]

26 Jun 2024

Effectiveness of D-dimer in predicting distant metastasis in colorectal cancer

PONE-D-23-43532R1

Dear Dr. Diao,

We’re pleased to inform you that your manuscript has been judged scientifically suitable for publication and will be formally accepted for publication once it meets all outstanding technical requirements.

Kind regards,

Raffaele Serra, M.D., Ph.D

Academic Editor

PLOS ONE

Additional Editor Comments (optional):

amended manuscript is acceptable.
---

## [Editor Report · Acceptance letter]

4 Jul 2024

PONE-D-23-43532R1 

PLOS ONE

Dear Dr. Diao, 

I'm pleased to inform you that your manuscript has been deemed suitable for publication in PLOS ONE. Congratulations! Your manuscript is now being handed over to our production team.

Kind regards, 

on behalf of

Prof. Raffaele Serra 

Academic Editor

PLOS ONE